# Deep Multimodal Fusion by Channel Exchanging

**Yikai Wang**[1], **Wenbing Huang**[1], **Fuchun Sun**[1†], **Tingyang Xu**[2], **Yu Rong**[2], **Junzhou Huang**[2]

[1]Beijing National Research Center for Information Science and Technology (BNRist),
State Key Lab on Intelligent Technology and Systems,
Department of Computer Science and Technology, Tsinghua University   [2]Tencent AI Lab

`wangyk17@mails.tsinghua.edu.cn, hwenbing@126.com, fcsun@tsinghua.edu.cn,`
`tingyangxu@tencent.com, yu.rong@hotmail.com, jzhuang@uta.edu`

## Abstract

Deep multimodal fusion by using multiple sources of data for classification or regression has exhibited a clear advantage over the unimodal counterpart on various applications. Yet, current methods including aggregation-based and alignment-based fusion are still inadequate in balancing the trade-off between inter-modal fusion and intra-modal processing, incurring a bottleneck of performance improvement. To this end, this paper proposes Channel-Exchanging-Network (CEN), a parameter-free multimodal fusion framework that dynamically exchanges channels between sub-networks of different modalities. Specifically, the channel exchanging process is self-guided by individual channel importance that is measured by the magnitude of Batch-Normalization (BN) scaling factor during training. The validity of such exchanging process is also guaranteed by sharing convolutional filters yet keeping separate BN layers across modalities, which, as an add-on benefit, allows our multimodal architecture to be almost as compact as a unimodal network. Extensive experiments on semantic segmentation via RGB-D data and image translation through multi-domain input verify the effectiveness of our CEN compared to current state-of-the-art methods. Detailed ablation studies have also been carried out, which provably affirm the advantage of each component we propose. Our code is available at `https://github.com/yikaiw/CEN`.

## 1   Introduction

Encouraged by the growing availability of low-cost sensors, *multimodal fusion* that takes advantage of data obtained from different sources/structures for classification or regression has become a central problem in machine learning [4]. Joining the success of deep learning, multimodal fusion is recently specified as *deep multimodal fusion* by introducing end-to-end neural integration of multiple modalities [37], and it has exhibited remarkable benefits against the unimodal paradigm in semantic segmentation [28, 44], action recognition [13, 14, 43], visual question answering [1, 22], and many others [3, 25, 51].

A variety of works have been done towards deep multimodal fusion [37]. Regarding the type of how they fuse, existing methods are generally categorized into *aggregation-based* fusion, *alignment-based* fusion, and the mixture of them [4]. The aggregation-based methods employ a certain operation (*e.g.* averaging [18], concatenation [34, 50], and self-attention [44]) to combine multimodal sub-networks into a single network. The alignment-based fusion [9, 43, 46], instead, adopts a regulation loss to align the embedding of all sub-networks while keeping full propagation for each of them. The difference between such two mechanisms is depicted in Figure 1. Another categorization of multimodal fusion can be specified as early, middle, and late fusion, depending on when to fuse, which have been discussed in earlier works [2, 7, 17, 41] and also in the deep learning literature [4, 26, 27, 45].

---

[†]Corresponding author: Fuchun Sun.

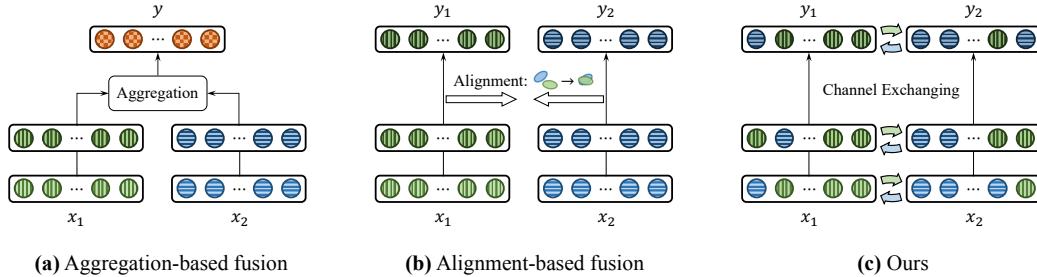

(a) Aggregation-based fusion     (b) Alignment-based fusion     (c) Ours

Figure 1: A sketched comparison between existing fusion methods and ours.

Albeit the fruitful progress, it remains a great challenge on how to integrate the common information across modalities, meanwhile preserving the specific patterns of each one. In particular, the aggregation-based fusion is prone to underestimating the intra-modal propagation once the multi-modal sub-networks have been aggregated. On the contrary, the alignment-based fusion maintains the intra-modal propagation, but it always delivers ineffective inter-modal fusion owing to the weak message exchanging by solely training the alignment loss. To balance between inter-modal fusion and intra-modal processing, current methods usually resort to careful hierarchical combination of the aggregation and alignment fusion for enhanced performance, at a cost of extra computation and engineering overhead [11, 28, 50].

**Present Work.** We propose Channel-Exchanging-Network (CEN) which is parameter-free, adaptive, and effective. Instead of using aggregation or alignment as before, CEN dynamically exchanges the channels between sub-networks for fusion (see Figure 1(c)). The core of CEN lies in its smaller-norm-less-informative assumption inspired from network pruning [32, 48]. To be specific, we utilize the scaling factor (*i.e.* $\gamma$) of Batch-Normalization (BN) [23] as the importance measurement of each corresponding channel, and replace the channels associated with close-to-zero factors of each modality with the mean of other modalities. Such message exchanging is parameter-free and self-adaptive, as it is dynamically controlled by the scaling factors that are determined by the training itself. Besides, we only allow directed channel exchanging within a certain range of channels in each modality to preserve intra-modal processing. More details are provided in § 3.3. Necessary theories on the validity of our idea are also presented in § 3.5.

Another hallmark of CEN is that the parameters except BN layers of all sub-networks are shared with each other (§ 3.4). Although this idea is previously studied in [8, 47], we apply it here to serve specific purposes in CEN: by using private BNs, as already discussed above, we can determine the channel importance for each individual modality; by sharing convolutional filters, the corresponding channels among different modalities are embedded with the same mapping, thus more capable of modeling the modality-common statistic. This design further compacts the multimodal architecture to be almost as small as the unimodal one.

We evaluate our CEN on two studies: semantic segmentation via RGB-D data [40, 42] and image translation through multi-domain input [49]. It demonstrates that CEN yields remarkably superior performance than various kinds of fusion methods based on aggregation or alignment under a fair condition of comparison. In terms of semantic segmentation particularly, our CEN significantly outperforms state-of-the-art methods on two popular benchmarks. We also conduct ablation studies to isolate the benefit of each proposed component. More specifications are provided in § 4.

## 2   Related Work

We introduce the methods of deep multimodal fusion, and the concepts related to our paper.

**Deep multimodal fusion.** As discussed in introduction, deep multimodal fusion methods can be mainly categorized into aggregation-based fusion and alignment-based fusion [4]. Due to the weakness in intra-modal processing, recent aggregation-based works perform feature fusion while still maintaining the sub-networks of all modalities [11, 29]. Besides, [18] points out the performance by fusion is highly affected by the choice of which layer to fuse. Alignment-based fusion methods align multimodal features by applying the similarity regulation, where Maximum-Mean-Discrepancy

(MMD) [15] is usually adopted for the measurement. However, simply focusing on unifying the whole distribution may overlook the specific patterns in each domain/modality [6, 43]. Hence, [46] provides a way that may alleviate this issue, which correlates modality-common features while simultaneously maintaining modality-specific information. There is also a portion of the multimodal learning literature based on modulation [10, 12, 45]. Different from these types of fusion methods, we propose a new fusion method by channel exchanging, which potentially enjoys the guarantee to both sufficient inter-model interactions and intra-modal learning.

**Other related concepts.** The idea of using BN scaling factor to evaluate the importance of CNN channels has been studied in network pruning [32, 48] and representation learning [39]. Moreover, [32] enforces $\ell_1$ norm penalty on the scaling factors and explicitly prunes out filters meeting a sparsity criteria. Here, we apply this idea as an adaptive tool to determine where to exchange and fuse. CBN [45] performs cross-modal message passing by modulating BN of one modality conditional on the other, which is clearly different from our method that directly exchanges channels between different modalities for fusion. ShuffleNet [52] proposes to shuffle a portion of channels among multiple groups for efficient propagation in light-weight networks, which is similar to our idea of exchanging channels for message fusion. Yet, while the motivation of our paper is highly different, the exchanging process is self-determined by the BN scaling factors, instead of the random exchanging in ShuffleNet.

# 3   Channel Exchanging Networks

In this section, we introduce our CEN, by mainly specifying its two fundamental components: the channel exchanging process and the sub-network sharing mechanism, followed by necessary analyses.

## 3.1   Problem Definition

Suppose we have the $i$-th input data of $M$ modalities, $\boldsymbol{x}^{(i)} = \{\boldsymbol{x}_m^{(i)} \in \mathbb{R}^{C \times (H \times W)}\}_{m=1}^M$, where $C$ denotes the number of channels, $H$ and $W$ denote the height and width of the feature map[2]. We define $N$ as the batch-size. The goal of deep multimodal fusion is to determine a multi-layer network $f(\boldsymbol{x}^{(i)})$ (particularly CNN in this paper) whose output $\hat{\boldsymbol{y}}^{(i)}$ is expected to fit the target $\boldsymbol{y}^{(i)}$ as much as possible. This can be implemented by minimizing the empirical loss as

$$\min_f \frac{1}{N} \sum_{i=1}^N \mathcal{L}\left(\hat{\boldsymbol{y}}^{(i)} = f(\boldsymbol{x}^{(i)}), \boldsymbol{y}^{(i)}\right). \tag{1}$$

We now introduce two typical kinds of instantiations to Equation 1:

**I.** The aggregation-based fusion first processes each $m$-th modality with a separate sub-network $f_m$ and then combine all their outputs via an aggregation operation followed by a global mapping. In formal, it computes the output by

$$\hat{\boldsymbol{y}}^{(i)} = f(\boldsymbol{x}^{(i)}) = h(\text{Agg}(f_1(\boldsymbol{x}_1^{(i)}), \cdots, f_M(\boldsymbol{x}_M^{(i)}))), \tag{2}$$

where $h$ is the global network and Agg is the aggregation function. The aggregation can be implemented as averaging [18], concatenation [50], and self-attention [44]. All networks are optimized via minimizing Equation 1.

**II.** The alignment-based fusion leverages an alignment loss for capturing the inter-modal concordance while keeping the outputs of all sub-networks $f_m$. Formally, it solves

$$\min_{f_{1:M}} \frac{1}{N} \sum_{i=1}^N \mathcal{L}\left(\sum_{m=1}^M \alpha_m f_m(\boldsymbol{x}_m^{(i)}), \boldsymbol{y}^{(i)}\right) + \text{Alig}_{f_{1:M}}(\boldsymbol{x}^{(i)}), \quad s.t. \sum_{m=1}^M \alpha_m = 1, \tag{3}$$

where the alignment $\text{Alig}_{f_{1:M}}$ is usually specified as Maximum-Mean-Discrepancy (MMD) [15] between certain hidden features of sub-networks, and the final output $\sum_{m=1}^M \alpha_m f_m(\boldsymbol{x}_m^{(i)})$ is an

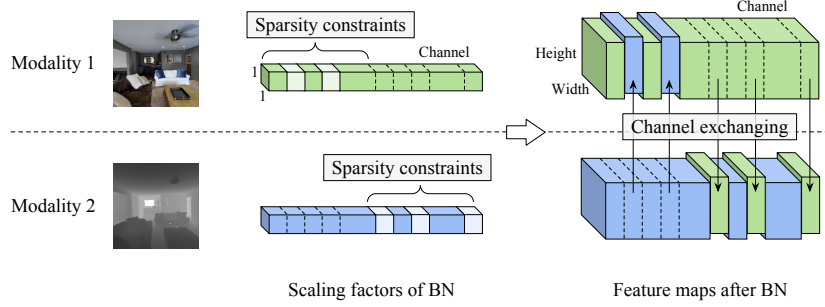

Figure 2: An illustration of our multimodal fusion strategy. The sparsity constraints on scaling factors are applied to disjoint regions of different modalities. A feature map will be replaced by that of other modalities at the same position, if its scaling factor is lower than a threshold.

ensemble of $f_m$ associated with the decision score $\alpha_m$ which is learnt by an additional softmax output to meet the simplex constraint.

As already discussed in introduction, both fusion methods are insufficient to determine the trade-off between fusing modality-common information and preserving modality-specific patterns. In contrast, our CEN is able to combine their best, the details of which are clarified in the next sub-section.

## 3.2 Overall Framework

The whole optimization objective of our method is

$$\min_{f_{1:M}} \frac{1}{N} \sum_{i=1}^{N} \mathcal{L} \left( \sum_{m=1}^{M} \alpha_m f_m(\boldsymbol{x}^{(i)}), \boldsymbol{y}^{(i)} \right) + \lambda \sum_{m=1}^{M} \sum_{l=1}^{L} |\hat{\boldsymbol{\gamma}}_{m,l}|, \quad s.t. \sum_{m=1}^{M} \alpha_m = 1, \qquad (4)$$

where,

- The sub-network $f_m(\boldsymbol{x}^{(i)})$ (opposed to $f_m(\boldsymbol{x}_m^{(i)})$ in Equation 3 of the alignment fusion) fuses multimodal information by channel exchanging, as we will detail in § 3.3;

- Each sub-network is equipped with BN layers containing the scaling factors $\gamma_{m,l}$ for the $l$-th layer, and we will penalize the $\ell_1$ norm of their certain portion $\hat{\gamma}_{m,l}$ for sparsity, which is presented in § 3.3;

- The sub-network $f_m$ shares the same parameters except BN layers to facilitate the channel exchanging as well as to compact the architecture further, as introduced in § 3.4;

- The decision scores of the ensemble output, $\alpha_m$, are trained by a softmax output similar to the alignment-based methods.

By the design of Equation 4, we conduct a parameter-free message fusion across modalities while maintaining the self-propagation of each sub-network so as to characterize the specific statistic of each modality. Moreover, our fusion of channel exchanging is self-adaptive and easily embedded to everywhere of the sub-networks, with the details given in what follows.

## 3.3 Channel Exchanging by Comparing BN Scaling Factor

Prior to introducing the channel exchanging process, we first review the BN layer [23], which is used widely in deep learning to eliminate covariate shift and improve generalization. We denote by $\boldsymbol{x}_{m,l}$ the $l$-th layer feature maps of the $m$-th sub-network, and by $\boldsymbol{x}_{m,l,c}$ the $c$-th channel. The BN layer performs a normalization of $\boldsymbol{x}_{m,l}$ followed by an affine transformation, namely,

$$\boldsymbol{x}'_{m,l,c} = \gamma_{m,l,c} \frac{\boldsymbol{x}_{m,l,c} - \mu_{m,l,c}}{\sqrt{\sigma_{m,l,c}^2 + \epsilon}} + \beta_{m,l,c}, \qquad (5)$$

where, $\mu_{m,l,c}$ and $\sigma_{m,l,c}$ compute the mean and the standard deviation, respectively, of all activations over all pixel locations ($H$ and $W$) for the current mini-batch data; $\gamma_{m,l,c}$ and $\beta_{m,l,c}$ are the trainable

scaling factor and offset, respectively; $\epsilon$ is a small constant to avoid divisions by zero. The $(l+1)$-th layer takes $\{\boldsymbol{x}'_{m,l,c}\}_c$ as input after a non-linear function.

The factor $\gamma_{m,l,c}$ in Equation 5 evaluates the correlation between the input $\boldsymbol{x}_{m,l,c}$ and the output $\boldsymbol{x}'_{m,l,c}$ during training. The gradient of the loss *w.r.t.* $\boldsymbol{x}_{m,l,c}$ will approach 0 if $\gamma_{m,l,c} \to 0$, implying that $\boldsymbol{x}_{m,l,c}$ will lose its influence to the final prediction and become redundant thereby. Moreover, we will prove in § 3.5 that the state of $\gamma_{m,l,c} = 0$ is attractive with a high probability, given the $\ell_1$ norm regulation in Equation 4. In other words, once the current channel $\boldsymbol{x}_{m,l,c}$ becomes redundant due to $\gamma_{m,l,c} \to 0$ at a certain training step, it will almost do henceforth.

It thus motivates us to replace the channels of small scaling factors with the ones of other sub-networks, since those channels potentially are redundant. To do so, we derive

$$\boldsymbol{x}'_{m,l,c} = \begin{cases} \gamma_{m,l,c} \dfrac{\boldsymbol{x}_{m,l,c} - \mu_{m,l,c}}{\sqrt{\sigma^2_{m,l,c} + \epsilon}} + \beta_{m,l,c}, & \text{if} \quad \gamma_{m,l,c} > \theta; \\ \dfrac{1}{M-1} \sum\limits_{m' \neq m}^{M} \gamma_{m',l,c} \dfrac{\boldsymbol{x}_{m',l,c} - \mu_{m',l,c}}{\sqrt{\sigma^2_{m',l,c} + \epsilon}} + \beta_{m',l,c}, & \text{else}; \end{cases} \quad (6)$$

where, the current channel is replaced with the mean of other channels if its scaling factor is smaller than a certain threshold $\theta \approx 0^+$. In a nutshell, if one channel of one modality has little impact to the final prediction, then we replace it with the mean of other modalities. We apply Equation 6 for each modality before feeding them into the nonlinear activation followed by the convolutions in the next layer. Gradients are detached from the replaced channel and back-propagated through the new ones.

In our implementation, we divide the whole channels into $M$ equal sub-parts, and only perform the channel exchanging in each different sub-part for different modality. We denote the scaling factors that are allowed to be replaced as $\hat{\boldsymbol{\gamma}}_{m,l}$. We further impose the sparsity constraint on $\hat{\boldsymbol{\gamma}}_{m,l}$ in Equation 4 to discover unnecessary channels. As the exchanging in Equation 6 is a directed process within only one sub-part of channels, it hopefully can not only retain modal-specific propagation in the other $M-1$ sub-parts but also avoid unavailing exchanging since $\gamma_{m',l,c}$, different from $\hat{\gamma}_{m,l,c}$, is out of the sparsity constraint. Figure 2 illustrates our channel exchanging process.

### 3.4 Sub-Network Sharing with Independent BN

It is known in [8, 47] that leveraging private BN layers is able to characterize the traits of different domains or modalities. In our method, specifically, different scaling factors (Equation 5) evaluate the importance of the channels of different modalities, and they should be decoupled.

With the exception of BN layers, all sub-networks $f_m$ share all parameters with each other including convolutional filters[3]. The hope is that we can further reduce the network complexity and therefore improve the predictive generalization. Rather, considering the specific design of our framework, sharing convolutional filters is able to capture the common patterns in different modalities, which is a crucial purpose of multimodal fusion. In our experiments, we conduct multimodal fusion on RGB-D images or on other domains of images corresponding to the same image content. In this scenario, all modalities are homogeneous in the sense that they are just different views of the same input. Thus, sharing parameters between different sub-networks still yields promisingly expressive power. Nevertheless, when we are dealing with heterogeneous modalities (*e.g.* images with text sequences), it would impede the expressive power of the sub-networks if keeping sharing their parameters, hence a more dexterous mechanism is suggested, the discussion of which is left for future exploration.

### 3.5 Analysis

**Theorem 1** *Suppose $\{\gamma_{m,l,c}\}_{m,l,c}$ are the BN scaling factors of any multimodal fusion network (without channel exchanging) optimized by Equation 4. Then the probability of $\gamma_{m,l,c}$ being attracted to $\gamma_{m,l,c} = 0$ during training* (a.k.a. $\gamma_{m,l,c} = 0$ is the local minimum) *is equal to $2\Phi(\lambda|\frac{\partial L}{\partial \boldsymbol{x}'_{m,l,c}}|^{-1}) - 1$, where $\Phi$ derives the cumulative probability of standard Gaussian.*

In practice, especially when approaching the convergence point, the magnitude of $\frac{\partial L}{\partial \boldsymbol{x}'_{m,l,c}}$ is usually very close to zero, indicating that the probability of staying around $\gamma_{m,l,c} = 0$ is large. In other words,

Table 1: Detailed results for different versions of our CEN on NYUDv2. All results are obtained with the backbone RefineNet (ResNet101) of single-scale evaluation for test.

| Convs | BNs | $\ell_1$ Regulation | Exchange | Mean IoU (%) | | |
|---|---|---|---|---|---|---|
| | | | | RGB | Depth | Ensemble |
| Unshared | Unshared | × | × | 45.5 | 35.8 | 47.6 |
| Shared | Shared | × | × | 43.7 | 35.5 | 45.2 |
| Shared | Unshared | × | × | 46.2 | 38.4 | 48.0 |
| Shared | Unshared | Half-channel | × | 46.0 | 38.1 | 47.7 |
| Shared | Unshared | Half-channel | ✓ | **49.7** | **45.1** | **51.1** |
| Shared | Unshared | All-channel | ✓ | 48.6 | 39.0 | 49.8 |

Table 2: Comparison with three typical fusion methods including concatenation (concat), fusion by alignment (align), and self-attention (self-att.) on NYUDv2. All results are obtained with the backbone RefineNet (ResNet101) of single-scale evaluation for test.

| Modality | Approach | Commonly-used setting | | Same with our setting | | Params used |
|---|---|---|---|---|---|---|
| | | Mean IoU (%) | Params in total (M) | Mean IoU (%) RGB / Depth / Ensemble | Params in total (M) | for fusion (M) |
| RGB | Uni-modal | 45.5 | 118.1 | 45.5 / - / - | 118.1 | - |
| Depth | Uni-modal | 35.8 | 118.1 | - / 35.8 / - | 118.1 | - |
| RGB-D | Concat (early) | 47.2 | 120.1 | 47.0 / 37.5 / 47.6 | 118.8 | 0.6 |
| | Concat (middle) | 46.7 | 147.7 | 46.6 / 37.0 / 47.4 | 120.3 | 2.1 |
| | Concat (late) | 46.3 | 169.0 | 46.3 / 37.2 / 46.9 | 126.6 | 8.4 |
| | Concat (all-stage) | 47.5 | 171.7 | 47.8 / 36.9 / 48.3 | 129.4 | 11.2 |
| | Align (early) | 46.4 | 238.8 | 46.3 / 35.8 / 46.7 | 120.8 | 2.6 |
| | Align (middle) | 47.9 | 246.7 | 47.7 / 36.0 / 48.1 | 128.7 | 10.5 |
| | Align (late) | 47.6 | 278.1 | 47.3 / 35.4 / 47.6 | 160.1 | 41.9 |
| | Align (all-stage) | 46.8 | 291.9 | 46.6 / 35.5 / 47.0 | 173.9 | 55.7 |
| | Self-att. (early) | 47.8 | 124.9 | 47.7 / 38.3 / 48.2 | 123.6 | 5.4 |
| | Self-att. (middle) | 48.3 | 166.9 | 48.0 / 38.1 / 48.7 | 139.4 | 21.2 |
| | Self-att. (late) | 47.5 | 245.5 | 47.6 / 38.1 / 48.3 | 203.2 | 84.9 |
| | Self-att. (all-stage) | 48.7 | 272.3 | 48.5 / 37.7 / 49.1 | 231.0 | 112.8 |
| | Ours | - | - | **49.7 / 45.1 / 51.1** | **118.2** | **0.0** |

when the scaling factor of one channel is equal to zero, this channel will almost become redundant during later training process, which will be verified by our experiment in the appendix. Therefore, replacing the channels of $\gamma_{m,l,c} = 0$ with other channels (or anything else) will only enhance the trainablity of the model. We immediately have the following corollary,

**Corollary 1** *If the minimal of Equation 4 implies $\gamma_{m,l,c} = 0$, then the channel exchanging by Equation 6 (assumed no crossmodal parameter sharing) will only decrease the training loss,* i.e. $\min_{f'_{1:M}} L \leq \min_{f_{1:M}} L$, *given the sufficiently expressive $f'_{1:M}$ and $f_{1:M}$ which denote the cases with and without channel exchanging, respectively.*

## 4 Experiments

We contrast the performance of CEN against existing multimodal fusion methods on two different tasks: semantic segmentation and image-to-image translation. The frameworks for both tasks are in the encoder-decoder style. Note that we only perform multimodal fusion within the encoders of different modalities throughout the experiments. Our codes are complied on PyTorch [35].

### 4.1 Semantic Segmentation

**Datasets.** We evaluate our method on two public datasets NYUDv2 [40] and SUN RGB-D [42], which consider RGB and depth as input. Regarding NYUDv2, we follow the standard settings and adopt the split of 795 images for training and 654 for testing, with predicting standard 40 classes [16]. SUN RGB-D is one of the most challenging large-scale benchmarks towards indoor semantic segmentation, containing 10,335 RGB-D images of 37 semantic classes. We use the public train-test split (5,285 vs 5,050).

**Implementation.** We consider RefineNet [31]/PSPNet [53] as our segmentation framework whose backbone is implemented by ResNet [19] pretrained from ImageNet dataset [38]. The initial learn-

Table 3: Comparison with SOTA methods on semantic segmentation.

| Modality | Approach | Backbone Network | NYUDv2 | | | SUN RGB-D | | |
|---|---|---|---|---|---|---|---|---|
| | | | Pixel Acc. (%) | Mean Acc. (%) | Mean IoU (%) | Pixel Acc. (%) | Mean Acc. (%) | Mean IoU (%) |
| RGB | FCN-32s [33] | VGG16 | 60.0 | 42.2 | 29.2 | 68.4 | 41.1 | 29.0 |
| | RefineNet [31] | ResNet101 | 73.8 | 58.8 | 46.4 | 80.8 | 57.3 | 46.3 |
| | RefineNet [31] | ResNet152 | 74.4 | 59.6 | 47.6 | 81.1 | 57.7 | 47.0 |
| RGB-D | FuseNet [18] | VGG16 | 68.1 | 50.4 | 37.9 | 76.3 | 48.3 | 37.3 |
| | ACNet [21] | ResNet50 | - | - | 48.3 | - | - | 48.1 |
| | SSMA [44] | ResNet50 | 75.2 | 60.5 | 48.7 | 81.0 | 58.1 | 45.7 |
| | SSMA [44] † | ResNet101 | 75.8 | 62.3 | 49.6 | 81.6 | 60.4 | 47.9 |
| | CBN [45] † | ResNet101 | 75.5 | 61.2 | 48.9 | 81.5 | 59.8 | 47.4 |
| | 3DGNN [36] | ResNet101 | - | - | - | - | 57.0 | 45.9 |
| | SCN [30] | ResNet152 | - | - | 49.6 | - | - | 50.7 |
| | CFN [29] | ResNet152 | - | - | 47.7 | - | - | 48.1 |
| | RDFNet [28] | ResNet101 | 75.6 | 62.2 | 49.1 | 80.9 | 59.6 | 47.2 |
| | RDFNet [28] | ResNet152 | 76.0 | 62.8 | 50.1 | 81.5 | 60.1 | 47.7 |
| | Ours-RefineNet (single-scale) | ResNet101 | 76.2 | 62.8 | 51.1 | 82.0 | 60.9 | 49.6 |
| | Ours-RefineNet | ResNet101 | 77.2 | 63.7 | 51.7 | 82.8 | 61.9 | 50.2 |
| | Ours-RefineNet | ResNet152 | 77.4 | 64.8 | 52.2 | 83.2 | 62.5 | 50.8 |
| | Ours-PSPNet | ResNet152 | **77.7** | **65.0** | **52.5** | **83.5** | **63.2** | **51.1** |

† indicates our implemented results.

ing rates are set to $5 \times 10^{-4}$ and $3 \times 10^{-3}$ for the encoder and decoder, respectively, both of which are reduced to their halves every 100/150 epochs (total epochs 300/450) on NYUDv2 with ResNet101/ResNet152 and every 20 epochs (total epochs 60) on SUN RGB-D. The mini-batch size, momentum and weight decay are selected as 6, 0.9, and $10^{-5}$, respectively, on both datasets. We set $\lambda = 5 \times 10^{-3}$ in Equation 4 and the threshold to $\theta = 2 \times 10^{-2}$ in Equation 6. Unless otherwise specified, we adopt the multi-scale strategy [28, 31] for test. We employ the Mean IoU along with Pixel Accuracy and Mean Accuracy as evaluation metrics following [31]. Full implementation details are referred to our appendix.

**The validity of each proposed component.** Table 1 summarizes the results of different variants of CEN on NYUDv2. We have the following observations: **1.** Compared to the unshared baseline, sharing the convolutional parameters greatly boosts the performance particularly on the Depth modality (35.8 vs 38.4). Yet, the performance will encounter a clear drop if we additionally share the BN layers. This observation is consistent with our analyses in § 3.4 due to the different roles of convolutional filters and BN parameters. **2.** After carrying out directed channel exchanging under the $\ell_1$ regulation, our model gains a huge improvement on both modalities, *i.e.* from 46.0 to 49.7 on RGB, and from 38.1 to 45.1 on Depth, and finally increases the ensemble Mean IoU from 47.6 to 51.1. It thus verifies the effectiveness of our proposed mechanism on this task. **3.** Note that the channel exchanging is only available on a certain portion of each layer (*i.e.* the half of the channels in the two-modal case). When we remove this constraint and allow all channels to be exchanged by Equation 6, the accuracy decreases, which we conjecture is owing to the detriment by impeding modal-specific propagation, if all channels are engaged in cross-modal fusion.

To further explain why channel exchanging works, Figure 3 displays the feature maps of RGB and Depth, where we find that the RGB channel with non-zero scaling factor mainly characterizes the texture, while the Depth channel with non-zero factor focuses more on the boundary; in this sense, performing channel exchanging can better combine the complementary properties of both modalities.

**Comparison with other fusion baselines.** Table 2 reports the comparison of our CEN with two aggregation-based methods: concatenation [50] and self-attention [44], and one alignment-based approach [46], using the same backbone. All baselines are implemented with the early, middle, late, and all stage fusion. Besides, for a more fair comparison, all baselines are further conducted under the same setting (except channel exchanging) with ours, namely, sharing convolutions with private BNs, and preserving the propagation of all sub-networks. Full details are provided in the appendix. It demonstrates that, on both settings, our method always outperforms others by an average improvement more than 2%. We also report the parameters used for fusion, *e.g.* the aggregation weights of two modalities in concatenation. While self-attention (all-stage) attains the closest performance to us (49.1 vs 51.1), the parameters it used for fusion are considerable, whereas our fusion is parameter-free.

**Comparison with SOTAs.** We contrast our method against a wide range of state-of-the-art methods. Their results are directly copied from previous papers if provided or re-implemented by us otherwise, with full specifications illustrated in the appendix. Table 3 concludes that our method equipped with PSPNet (ResNet152) achieves new records remarkably superior to previous methods in terms of all

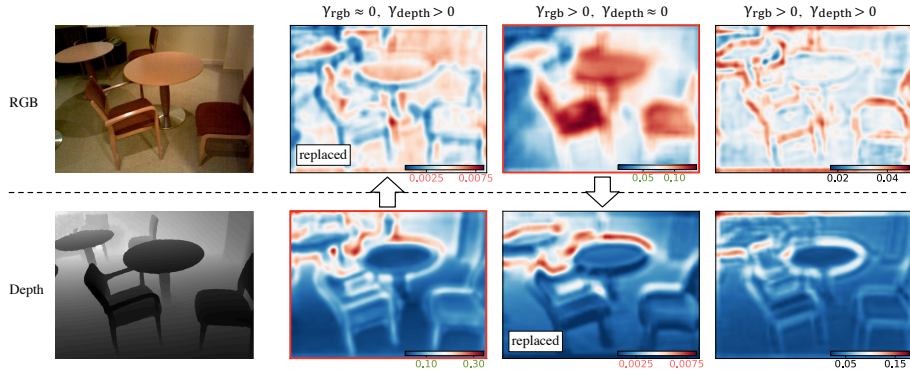

Figure 3: Visualization of the averaged feature maps for RGB and Depth. From left to right: the input images, the channels of $(\gamma_{rgb} \approx 0, \gamma_{depth} > 0)$, $(\gamma_{rgb} > 0, \gamma_{depth} \approx 0)$, and $(\gamma_{rgb} > 0, \gamma_{depth} > 0)$.

Table 4: Comparison on image-to-image translation. Evaluation metrics are FID/KID ($\times 10^{-2}$). Lower values indicate better performance.

| Modality | Ours | Baseline | Early | Middle | Late | All-layer |
|---|---|---|---|---|---|---|
| Shade+Texture →RGB | **62.63 / 1.65** | Concat | 87.46 / 3.64 | 95.16 / 4.67 | 122.47 / 6.56 | 78.82 / 3.13 |
| | | Average | 93.72 / 4.22 | 93.91 / 4.27 | 126.74 / 7.10 | 80.64 / 3.24 |
| | | Align | 99.68 / 4.93 | 95.52 / 4.75 | 98.33 / 4.70 | 92.30 / 4.20 |
| | | Self-att. | 83.60 / 3.38 | 90.79 / 3.92 | 105.62 / 5.42 | 73.87 / 2.46 |
| Depth+Normal →RGB | **84.33 / 2.70** | Concat | 105.17 / 5.15 | 100.29 / 3.37 | 116.51 / 5.74 | 99.08 / 4.28 |
| | | Average | 109.25 / 5.50 | 104.95 / 4.98 | 122.42 / 6.76 | 99.63 / 4.41 |
| | | Align | 111.65 / 5.53 | 108.92 / 5.26 | 105.85 / 4.98 | 105.03 / 4.91 |
| | | Self-att. | 100.70 / 4.47 | 98.63 / 4.35 | 108.02 / 5.09 | 96.73 / 3.95 |

Table 5: Multimodal fusion on image translation (to RGB) with modalities from 1 to 4.

| Modality | Depth | Normal | Texture | Shade | Depth+Normal | Depth+Normal +Texture | Depth+Normal +Texture+Shade |
|---|---|---|---|---|---|---|---|
| FID | 113.91 | 108.20 | 97.51 | 100.96 | 84.33 | 60.90 | 57.19 |
| KID ($\times 10^{-2}$) | 5.68 | 5.42 | 4.82 | 5.17 | 2.70 | 1.56 | 1.33 |

metrics on both datasets. In particular, given the same backbone, our method are still much better than RDFNet [28]. To isolate the contribution of RefineNet in our method, Table 3 also provides the uni-modal results, where we observe a clear advantage of multimodal fusion.

**Additional ablation studies.** In this part, we provide some additional experiments on NYUDv2, with RefineNet (ResNet101). Results are obtained with single-scale evaluation. **1.** As $\ell_1$ enables the discovery of unnecessary channels and comes as a pre-condition of Theorem 1, naively exchanging channels with a fixed portion (without using $\ell_1$ and threshold) could not reach good performance. For example, exchanging a fixed portion of 30% channels only gets IoU 47.2. We also find by only exchanging 30% channels at each down-sampling stage of the encoder, instead of every $3 \times 3$ convolutional layer throughout the encoder (like our CEN), the result becomes 48.6, which is much lower than our CEN (51.1). **2.** In Table 3, we provide results of our implemented CBN [45] by modulating the BN of depth conditional on RGB. The IoUs of CBN with unshared and shared convolutional parameters are 48.3 and 48.9, respectively. **3.** Directly summing activations (discarding the 1st term in Equation 6) results in IoU 48.1, which could reach 48.4 when summing with a learnt soft gate. **4.** If we replace the ensemble of expert with a concat-fusion block, the result will slightly reduce from 51.1 to 50.8. **5.** Besides, we try to exchange channels randomly like ShuffleNet or directly discard unimportant channels without channel exchanging, the IoUs of which are 46.8 and 47.5, respectively. All above ablations support the optimal design of our architecture.

## 4.2 Image-to-Image Translation

**Datasets.** We adopt Taskonomy [49], a dataset with 4 million images of indoor scenes of about 600 buildings. Each image in Taskonomy has more than 10 multimodal representations, including depth

(euclidean/zbuffer), shade, normal, texture, edge, principal curvature, etc. For efficiency, we sample 1,000 high-quality multimodal images for training, and 500 for validation.

**Implementation.** Following Pix2pix [24], we adopt the U-Net-256 structure for image translation with the consistent setups with [24]. The BN computations are replaced with Instance Normalization layers (INs), and our method (Equation 6) is still applicable. We adopt individual INs in the encoder, and share all other parameters including INs in the decoder. We set $\lambda$ to $10^{-3}$ for sparsity constraints and the threshold $\theta$ to $10^{-2}$. We adopt FID [20] and KID [5] as evaluation metrics, which will be introduced in our appendix.

**Comparison with other fusion baselines.** In Table 4, we evaluate the performance on two specific translation cases, *i.e.* Shade+Texture→RGB and Depth+Normal→RGB, with more examples included in the appendix. In addition to the three baselines used in semantic segmentation (Concat, Self-attention, Align), we conduct an extra aggregation-based method by using the average operation. All baselines perform fusion under 4 different kinds of strategies: early (at the 1st conv-layer), middle (the 4th conv-layer), late (the 8th conv-layer), and all-layer fusion. As shown in Table 4, our method yields much lower FID/KID than others, which supports the benefit of our proposed idea once again.

**Considering more modalities.** We now test whether our method is applicable to the case with more than 2 modalities. For this purpose, Table 5 presents the results of image translation to RGB by inputting from 1 to 4 modalities of Depth, Normal, Texture, and Shade. It is observed that increasing the number of modalities improves the performance consistently, suggesting much potential of applying our method towards various cases.

## 5    Conclusion

In this work, we propose Channel-Exchanging-Network (CEN), a novel framework for deep multimodal fusion, which differs greatly with existing aggregation-based and alignment-based multimodal fusion. The motivation behind is to boost inter-modal fusion while simultaneously keeping sufficient intra-modal processing. The channel exchanging is self-guided by channel importance measured by individual BNs, making our framework self-adaptive and compact. Extensive evaluations verify the effectiveness of our method.

## Acknowledgement

This work is jointly funded by National Natural Science Foundation of China and German Research Foundation (NSFC 61621136008/DFG TRR-169) in project "Crossmodal Learning" II, Tencent AI Lab Rhino-Bird Visiting Scholars Program (VS202006), and China Postdoctoral Science Foundation (Grant No.2020M670337).

## Broader Impact

This research enables fusing complementary information from different modalities effectively, which helps improve performance for autonomous vehicles and indoor manipulation robots, also making them more robust to environmental conditions, *e.g.* light, weather. Besides, instead of carefully designing hierarchical fusion strategies in existing methods, a global criterion is applied in our work for guiding multimodal fusion, which allows easier model deployment for practical applications. A drawback of bringing deep neural networks into multimodal fusion is its insufficient interpretability.

## Footnotes

[2]Although our paper is specifically interested in image data, our method is still general to other domains; for example, we can set $H = W = 1$ for vectors.

[3]If the input channels of different modalities are different (*e.g.* RGB and depth), we will broaden their sizes to be the same as their Least Common Multiple (LCM).

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
