[Supplementary Material]

# Supplementary Material of Deep Multimodal Fusion by Channel Exchanging

**Yikai Wang**[1], **Wenbing Huang**[1], **Fuchun Sun**[1], **Tingyang Xu**[2], **Yu Rong**[2], **Junzhou Huang**[2]

[1]Beijing National Research Center for Information Science and Technology (BNRist),
State Key Lab on Intelligent Technology and Systems,
Department of Computer Science and Technology, Tsinghua University  [2]Tencent AI Lab

wangyk17@mails.tsinghua.edu.cn, hwenbing@126.com, fcsun@tsinghua.edu.cn,
tingyangxu@tencent.com, yu.rong@hotmail.com, jzhuang@uta.edu

## 1 Proofs

**Theorem 1.** *Suppose $\{\gamma_{m,l,c}\}_{m,l,c}$ are the BN scaling factors of any multimodal fusion network (without channel exchanging) optimized by Equation 4. Then the probability of $\gamma_{m,l,c}$ being attracted to $\gamma_{m,l,c} = 0$ during training* (a.k.a. $\gamma_{m,l,c} = 0$ is the local minimum) *is equal to $2\Phi(\lambda|\frac{\partial L}{\partial \boldsymbol{x}'_{m,l,c}}|^{-1}) - 1$, where $\Phi$ derives the cumulative probability of standard Gaussian.*

*Proof.* The proof is straightforward, since the gradient of $L$ w.r.t. $\gamma_{m,l,c}$ is $\frac{\partial L}{\partial \boldsymbol{x}'_{m,l,c}} \frac{\boldsymbol{x}_{m,l,c}-\mu_{m,l,c}}{\sqrt{\sigma^2_{m,l,c}+\epsilon}} + \lambda$ when $\gamma_{m,l,c} > 0$, or $\frac{\partial L}{\partial \boldsymbol{x}'_{m,l,c}} \frac{\boldsymbol{x}_{m,l,c}-\mu_{m,l,c}}{\sqrt{\sigma^2_{m,l,c}+\epsilon}} - \lambda$ when $\gamma_{m,l,c} < 0$[4], according to the BN definition in Equation 5 and the $\ell_1$ norm in Equation 4. Staying around $\gamma_{m,l,c} = 0$ during training implies that $\frac{\partial L}{\partial \boldsymbol{x}'_{m,l,c}} \frac{\boldsymbol{x}_{m,l,c}-\mu_{m,l,c}}{\sqrt{\sigma^2_{m,l,c}+\epsilon}} + \lambda > 0$ as well as $\frac{\partial L}{\partial \boldsymbol{x}'_{m,l,c}} \frac{\boldsymbol{x}_{m,l,c}-\mu_{m,l,c}}{\sqrt{\sigma^2_{m,l,c}+\epsilon}} - \lambda < 0$, the probability of which is $2\Phi(\lambda|\frac{\partial L}{\partial \boldsymbol{x}'_{m,l,c}}|^{-1}) - 1$ given that the quantity $\frac{\boldsymbol{x}_{m,l,c}-\mu_{m,l,c}}{\sqrt{\sigma^2_{m,l,c}+\epsilon}}$ can be considered as a random variable of standard Gaussian according to the central limit theorem. $\square$

Figure 4: Illustration of the conclusion by Theorem 1.

**Corollary 1.** *If the minimal of Equation 4 implies $\gamma_{m,l,c} = 0$, then the channel exchanging by Equation 6 (assumed no crossmodal parameter sharing) will only decrease the training loss,* i.e.

(a) Scaling factors of the first 128 channels (with sparsity constraints) when channel exchanging is applied

(b) Scaling factors of the first 128 channels (with sparsity constraints) when channel exchanging is **NOT** applied

Figure 5: We plot BN scaling factors with sparsity constraints vs training steps. We observe that whether using channel exchanging or not, $\gamma$ that closes to zero can hardly recover, which verifies our conjecture in Theorem 1. The experiment is conducted on NYUDv2 with RefineNet (ResNet101). We choose the 8th layer of convolutional layers that have $3 \times 3$ kernels, and there are totally 256 channels in this layer. Regarding the RGB modality, the sparsity constraints to BN scaling factors are applied for the first 128 channels.

$\min_{f'_{1:M}} L \leq \min_{f_{1:M}} L$, *given the sufficiently expressive $f'_{1:M}$ and $f_{1:M}$ which denote the cases with and without channel exchanging, respectively.*

*Proof.* We only need to prove for any $f_{1:M}$, we can design a specific $f'_{1:M}$ that shares the same output as $f_{1:M}$ if $\gamma_{m,l,c} = 0$.

- In $f_{1:M}$, the BN layer is followed by a ReLU function and a convolutional layer. We suppose the following convolutional weight for the $c$-th input channel $\boldsymbol{x}'_{m,l,c}$ is $\boldsymbol{W}_{m,l+1,c}$ and the bias is $b_{m,l+1}$. Thus, the quantity related to $\boldsymbol{x}'_{m,l,c}$ in the $(l+1)$-th layer is $\boldsymbol{W}_{m,l+1,c} \otimes \sigma(\boldsymbol{x}'_{m,l,c}) + b_{m,l+1}$, where $\otimes$ denotes the convolution operation and $\sigma$ is the ReLU function. Since $\gamma_{m,l,c} = 0$, this term can be translated as $\boldsymbol{W}_{m,l+1,c} \otimes \sigma(\beta_{m,l,c}) + b_{m,l+1}$, which is a constant feature map.

- As for $f'_{1:M}$, we apply the similar denotations, and attain the term related to $\boldsymbol{x}'_{m,l,c}$ in the $(l+1)$-th layer as $\boldsymbol{W}'_{m,l+1,c} \otimes \sigma(\boldsymbol{x}'_{m,l,c}) + b'_{m,l+1}$.

By setting $b'_{m,l+1} = \boldsymbol{W}_{m,l+1,c} \otimes \sigma(\beta_{m,l,c}) + b_{m,l+1}$ and $\boldsymbol{W}'_{m,l+1,c} = 0$, we will always have $f'_{1:M} = f_{1:M}$, which concludes the proof. □

In Figure 4, we provide an illustration of the conclusion by Theorem 1. In Figure 5, we provide experimental results to verify our conjecture in Theorem 1, *i.e.* when the scaling factor of one channel is equal to zero at a certain training step, this channel will almost become redundant during later training process.

In summary, we know that $\ell_1$ makes the parameters sparse, but it can not tell if each sparse parameter will keep small in training considering the gradient in Equation 4 of our main paper. Conditional on BN, Theorem 1 proves that $\gamma = 0$ is attractive. Corollary 1 states that $f'$ is more expressive than $f$ when $\gamma = 0$, and thus the optimal $f'$ always outputs no higher loss, which, yet, is not true for arbitrary $f'$ (*e.g.* $f' = 10^6$). Besides, as stated in our main paper, Corollary 1 holds upon unshared convolutional parameters, and is consistent with Table 7 in the unshared scenario (full-channel: 49.1 vs half-channel: 48.5), although full-channel exchanging is worse under the sharing setting.

## 2 Implementation Details

In our experiments, we adopt ResNet101, ResNet152 for semantic segmentation and U-Net-256 for image-to-image translation. Regarding both ResNet structures, we apply sparsity constraints on Batch-Normalization (BN) scaling factors *w.r.t.* each convolutional layer (conv) with $3 \times 3$ kernels. These scaling factors further guide the channel exchanging process that exchanges a portion of feature maps after BN. For the conv layer with $7 \times 7$ kernels at the beginning of ResNet, and all other conv layers with $1 \times 1$ kernels, we do not apply sparsity constraints or channel exchanging. For U-Net, we apply sparsity constraints on Instance-Normalization (IN) scaling factors *w.r.t.* all conv layers (eight layers in total) in the encoder of the generator, and each is followed by channel exchanging.

We mainly use three multimodal fusion baselines in our paper, including concatenation, alignment and self-attention. Regarding the concatenation method, we stack multimodal feature maps along the channel, and then add a $1 \times 1$ convolutional layer to reduce the number of channels back to the original number. The alignment fusion method is a re-implementation of [5], and we follow its default settings for hyper-parameter, *e.g.* using 11 kernel functions for the multiple kernel Maximum Mean Discrepancy. The self-attention method is a re-implementation of the SSMA block proposed in [4], where we also follow the default settings, *e.g.* setting the channel reduction ratio $\eta$ to 16.

In Table 2 of our main paper, we adopt early, middle, late and all-stage fusion for each baseline method. In ResNet101, there are four stages with 3, 4, 23, 3 blocks, respectively. The early fusion, middle fusion and late fusion refer to fusing after the 2nd stage, 3rd stage and 4th stage respectively. All-stage fusion refers to fusing after the four stages.

We use a NVIDIA Tesla V100 with 32GB for the experiments.

We now introduce the metrics used in our image-to-image translation task. In Table 4 of our main paper, we adopt the following evaluation metrics:

Fréchet-Inception-Distance (FID) proposed by [3], contrasts the statistics of generated samples against real samples. The FID fits a Gaussian distribution to the hidden activations of InceptionNet for each compared image set and then computes the Fréchet distance (also known as the Wasserstein-2 distance) between those Gaussians. Lower FID is better, corresponding to generated images more similar to the real.

Kernel-Inception-Distance (KID) developed by [1], is a metric similar to the FID but uses the squared Maximum-Mean-Discrepancy (MMD) between Inception representations with a polynomial kernel.Unlike FID, KID has a simple unbiased estimator, making it more reliable especially when there are much more inception features channels than image numbers. Lower KID indicates more visual similarity between real and generated images. Regarding our implementation of KID, the hidden representations are derived from the Inception-v3 pool3 layer.

## 3 Additional Results

We provide three more image translation cases in Table 6, including RGB+Shade→Normal, RGB+Normal→Shade and RGB+Edge→Depth. For baseline methods, we adopt the same settings with Table 4 in our main paper, by adopting early (at the 1st conv-layer), middle (the 4th conv-layer), late (the 8th conv-layer) and all-layer fusion. We adopt MAE (L1 loss) and MSE (L2 loss) as evaluation metrics, and lower values indicate better performance. Our method yields lower MAE and MSE than baseline methods.

## 4 Results Visualization

In Figure 6 and Figure 7, we provide results visualization for the semantic segmentation task. We choose three baselines including concatenation (concat), alignment (align) and self-attention (self-att.). Among them, concatenation and self-attention methods adopt all-stage fusion, and the alignment method adopts middle fusion (fusion at the end of the 2nd ResNet stage).

In Figure 8, Figure 9 and Figure 10, we provide results visualization for the image translation task. Regarding this task, concatenation and self-attention methods adopt all-layer fusion (fusion at all

Table 6: Comparison on image-to-image translation. Evaluation metrics adopted are MAE ($\times 10^{-1}$)/MSE ($\times 10^{-1}$). Lower values indicate better performance.

| Modality | Ours | Baseline | Early | Middle | Late | All-layer |
|---|---|---|---|---|---|---|
| RGB+Shade →Normal | **1.12 / 2.51** | Concat | 1.33 / 2.83 | 1.22 / 2.65 | 1.39 / 2.88 | 1.34 / 2.85 |
| | | Average | 1.42 / 3.05 | 1.26 / 2.70 | 1.40 / 2.90 | 1.28 / 2.83 |
| | | Align | 1.45 / 3.11 | 1.39 / 2.93 | 1.28 / 2.76 | 1.52 / 3.25 |
| | | Self-att. | 1.30 / 2.82 | 1.18 / 2.59 | 1.42 / 2.91 | 1.26 / 2.76 |
| RGB+Normal →Shade | **1.10 / 1.72** | Concat | 1.56 / 2.45 | 1.38 / 2.12 | 1.26 / 1.92 | 1.28 / 2.02 |
| | | Average | 1.46 / 2.29 | 1.28 / 2.04 | 1.51 / 2.39 | 1.23 / 1.86 |
| | | Align | 1.39 / 2.26 | 1.32 / 2.16 | 1.27 / 2.04 | 1.41 / 2.21 |
| | | Self-att. | 1.21 / 1.83 | 1.15 / 1.73 | 1.45 / 2.28 | 1.18 / 1.76 |
| RGB+Edge →Depth | **0.28 / 0.66** | Concat | 0.34 / 0.75 | 0.32 / 0.74 | 0.38 / 0.79 | 0.33 / 0.75 |
| | | Average | 0.36 / 0.78 | 0.34 / 0.76 | 0.36 / 0.77 | 0.33 / 0.74 |
| | | Align | 0.44 / 0.89 | 0.39 / 0.82 | 0.42 / 0.86 | 0.44 / 0.90 |
| | | Self-att. | 0.30 / 0.71 | 0.33 / 0.73 | 0.34 / 0.75 | 0.30 / 0.70 |

eight layers in the encoder), and the alignment method adopts middle fusion (fusion at the 4th layer). We adopt these settings in order to achieve high performance for each baseline method.

In the captions of these figures, we detail the prediction difference of different methods.

## 5 Ablation Studies

In Table 7, we provide more cases as a supplement to Table 1 of our main paper. Specifically, we compare the results of channel exchaging when using shared/unshared conv parameters. According to these results, we believe our method is generally useful and channels are aligned to some extent even under the unshared setting.

In Table 8, we verify that sharing convolutional layers (convs) but using individual Instance-Normalization layers (INs) allows 2∼4 modalities trained in a single network, achieving even better performance than training with individual networks. Again, if we further sharing INs, there will be an obvious performance drop. More detailed comparison is provided in Table 9.

For the experiment Shade+Texture+Depth→RGB with shared convs and unshared INs, in Figure 11, we plot the proportion of IN scaling factors at the 7th conv layer in the encoder of U-Net. We compare the scaling factors when no sparsity constraints, sparsity constraints applied on all channels, and sparsity constraints applied on disjoint channels. In Figure 12, we further compare scaling factors on all conv layers. In Figure 13, we provide sensitivity analysis for $\lambda$ and $\theta$.

Table 7: Supplement to Table 1 in our main paper, with more cases. Detailed results for different versions of our CEN on NYUDv2. All results are obtained with the backbone RefineNet (ResNet101) of single-scale evaluation for test. We observe that sharing convs (with unshared BNs) results in better performance for our method.

| Convs | BNs | $\ell_1$ Regulation | Exchange | Mean IoU (%) | | |
|---|---|---|---|---|---|---|
| | | | | RGB | Depth | Ensemble |
| Unshared | Unshared | × | × | 45.5 | 35.8 | 47.6 |
| Shared | Shared | × | × | 43.7 | 35.5 | 45.2 |
| Shared | Unshared | × | × | 46.2 | 38.4 | 48.0 |
| Unshared | Unshared | Half-channel | × | 45.1 | 35.5 | 47.3 |
| Unshared | Unshared | Half-channel | ✓ | 46.5 | 41.6 | 48.5 |
| Shared | Unshared | Half-channel | × | 46.0 | 38.1 | 47.7 |
| Shared | Unshared | Half-channel | ✓ | **49.7** | **45.1** | **51.1** |
| Unshared | Unshared | All-channel | × | 44.6 | 35.3 | 46.6 |
| Unshared | Unshared | All-channel | ✓ | 46.8 | 41.7 | 49.1 |
| Shared | Unshared | All-channel | × | 46.1 | 37.9 | 47.5 |
| Shared | Unshared | All-channel | ✓ | 48.6 | 39.0 | 49.8 |

## Footnotes

[4]Here, we denote $\frac{\partial L}{\partial \boldsymbol{x}'_{m,l,c}} \frac{\boldsymbol{x}_{m,l,c}-\mu_{m,l,c}}{\sqrt{\sigma^2_{m,l,c}+\epsilon}} = \sum_{(i,j)=1}^{(H,W)} \frac{\partial L}{\partial \boldsymbol{x}'_{m,l,c}}(i,j) \frac{\boldsymbol{x}_{m,l,c}(i,j)-\mu_{m,l,c}}{\sqrt{\sigma^2_{m,l,c}+\epsilon}}$ for alleviation, where $i, j$ range over each pixel in $\boldsymbol{x}'_{m,l,c}$ or $\boldsymbol{x}_{m,l,c}$.

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

Figure 6: Visualization results for semantic segmentation. Images are collected from NYUDv2 and SUN RGB-D dataset. All results are obtained with the backbone RefineNet (ResNet101) of single-scale evaluation for test. We choose tough images where a number of tables and chairs need to be predicted. Besides, we compare segmentation results on images with low/high light intensity. we observe that the concatenation method is more sensitive to noises of the depth input (see the window at bottom line). Both concatenation and self-attention methods are weak in predicting thin objects *e.g.* table legs and chair legs. These objects are usually missed in the depth input, which may disturb the prediction results during fusion. Compared to baseline fusion methods, the prediction results of our method preserve more details, and are more robust to the light intensity.

Figure 7: Visualization results for semantic segmentation on Cityscapes dataset [2]. All results are obtained with the backbone PSPNet (ResNet101) of single-scale evaluation for test. Cityscapes is an outdoor dataset containing images from 27 cities in Germany and neighboring countries. The dataset contains 2,975 training, 500 validation and 1,525 test images. There are 20,000 additional coarse annotations provided by the dataset, which are not used for training in our experiments. For the baseline methods, we use white frames to highlight the regions with poor prediction results. We can observe that when the light intensity is high, the baseline methods are weak in capturing the boundary between the sky and buildings using the depth information. Besides, the concatenation and self-attention methods do not preserve fine-grained objects, *e.g.* traffic signs, and are sensitive to noises of the depth input (see the rightmost vehicle in the first group). In contrast, the prediction of our method are better at these aforementioned aspects.

Figure 8: Two groups of results comparison for image translation from Texture and Shade to RGB. We observe that the prediction solely predicted from the texture is vague at boundary lines, while the prediction from the shade misses some opponents, *e.g.* the pendant lamp, and is weak in predicting handrails. When fusing the two modalities, the concatenation method is uncertain at the regions where both modalities have disagreements. Alignment and self-attention are still weak in combining both modalities at details. Our results are clear at boundaries and fine-grained details.

Table 8: We compare training multimodal features in a parallel manner with different parameter sharing settings. Results of the proposed fusion method are reported at the last column. Evaluation metrics are FID/KID ($\times 10^{-2}$). We observe that the convolutional layers can be shared as long as we leave individual INs for different modalities, achieving even better performance.

| Modality | Network stream | Unshared convs unshared INs | Shared convs shared INs | Shared convs unshared INs | Multi-modal fusion |
|---|---|---|---|---|---|
| Shade+Texture →RGB | Shade | 102.21 / 5.25 | 112.40 / 5.58 | 100.69 / 4.51 | 72.07 / 2.32 |
| | Texture | 98.19 / 4.83 | 102.28 / 5.22 | 93.40 / 4.18 | 65.60 / 1.82 |
| | Ensemble | 92.72 / 4.15 | 96.31 / 4.36 | 87.91 / 3.73 | 62.63 / 1.65 |
| Shade+Texture +Depth →RGB | Shade | 101.86 / 5.18 | 115.51 / 5.77 | 98.49 / 4.07 | 69.37 / 2.21 |
| | Texture | 98.60 / 4.89 | 104.39 / 4.54 | 95.87 / 4.27 | 64.70 / 1.73 |
| | Depth | 114.18 / 5.71 | 121.40 / 6.23 | 107.07 / 5.19 | 71.61 / 2.27 |
| | Ensemble | 91.30 / 3.92 | 100.41 / 4.73 | 84.39 / 3.45 | 58.35 / 1.42 |
| Shade+Texture +Depth+Normal →RGB | Shade | 100.83 / 5.06 | 131.74 / 7.48 | 96.98 / 4.23 | 68.70 / 2.14 |
| | Texture | 97.34 / 4.77 | 109.45 / 4.86 | 94.64 / 4.22 | 63.26 / 1.69 |
| | Depth | 114.50 / 5.83 | 125.54 / 6.48 | 109.93 / 5.41 | 70.47 / 2.09 |
| | Normal | 108.65 / 5.45 | 113.15 / 5.72 | 99.38 / 4.45 | 67.73 / 1.98 |
| | Ensemble | 89.52 / 3.80 | 102.78 / 4.67 | 86.76 / 3.63 | 57.19 / 1.33 |

Figure 9: Two groups of results comparison for image translation from RGB and Edge to Depth. It is straightforward to find the benefits of multimodal fusion in this figure. The depth predicted by RGB is good at predicting numerical values, but is weak in capturing boundaries, which results in curving walls. Oppositely, the depth predicted by the edge well captures boundaries, but is weak in determining numerical values. The alignment fusion method is still weak in capturing boundaries. Both concatenation and self-attention methods are able to combine the advantages of both modalities, but the numerical values are still obviously lower than the ground truth. Our prediction achieves better performance compared to baseline methods.

Table 9: An Instance-Normalization layer consists of four components, including scaling factors $\gamma$, offsets $\beta$, running mean $\mu$ and variance $\sigma^2$. Following Table 8, we further compare the evaluation results when using unshared $\gamma, \beta$ only, and using unshared $\mu, \sigma^2$ only. Evaluation metrics are FID/KID ($\times 10^{-2}$). We observe these four components of INs are all essential to be unshared. Besides, using unshared scaling factors and offsets seems to be more important.

| Modality | Network stream | Unshared convs unshared INs | Shared convs unshared INs | Shared convs,$\gamma,\beta$ unshared $\mu,\sigma^2$ | Shared convs,$\mu,\sigma^2$ unshared $\gamma,\beta$ |
|---|---|---|---|---|---|
| Shade+Texture +Depth $\rightarrow$RGB | Shade | 101.86 / 5.18 | 98.49 / 4.07 | 107.86 / 5.53 | 105.29 / 5.29 |
| | Texture | 98.60 / 4.89 | 95.87 / 4.27 | 105.46 / 5.25 | 102.90 / 5.06 |
| | Depth | 114.18 / 5.71 | 102.07 / 4.89 | 118.35 / 6.07 | 114.35 / 5.80 |
| | Ensemble | 91.30 / 3.92 | 84.39 / 3.45 | 96.30 / 4.41 | 92.25 / 4.02 |
| Shade+Texture +Depth+Normal $\rightarrow$RGB | Shade | 100.83 / 5.06 | 96.98 / 4.23 | 113.56 / 5.65 | 102.74 / 5.17 |
| | Texture | 97.34 / 4.77 | 94.64 / 4.22 | 105.36 / 5.32 | 97.53 / 4.56 |
| | Depth | 114.50 / 5.83 | 109.93 / 5.41 | 119.31 / 6.20 | 112.73 / 5.60 |
| | Normal | 108.65 / 5.45 | 99.38 / 4.45 | 108.01 / 5.06 | 100.34 / 4.53 |
| | Ensemble | 89.52 / 3.80 | 86.76 / 3.63 | 95.56 / 4.64 | 89.26 / 3.91 |

Figure 10: Results comparison for image translation from RGB and Shade to Normal (upper group), and from RGB and Normal to Shade (lower group). Our fusion method again outperforms the other methods regarding both overall performance and details.

Figure 11: We use shared convs and unshared INs, and plot the proportion of scaling factors for each modality, at the 7th conv layer, *i.e.* $\gamma_c^{m,l,c}/(\gamma_c^{1,l,c} + \gamma_c^{2,l,c} + \gamma_c^{3,l,c})$, where $m = 1, 2, 3$ corresponding to Shade, Texture and Depth respectively, and $l = 7$. **Top**: no sparsity constraints are applied, where the scaling factor of each modality occupies a certain proportion at each channel. **Middle**: sparsity constraints are applied to all channels, where scaling factors of one modality could occupy a large proportion, indicating the channels are re-allocated to different modalities under the sparsity constraints. Yet this setting is not very suitable for channel exchanging, as a redundant feature map of one modality may be replaced by another redundant feature map. **Bottom**: sparsity constraints are applied to disjoint channels, which is our default setting.

Figure 12: Proportion of scaling factors in the U-Net encoder. We provide results at all layers. **Upper left**: no sparsity constraints are applied; **Upper right**: sparsity constraints are applied on all channels; **Bottom left**: sparsity constraints are applied on disjoint channels.

Figure 13: Sensitivity analysis for $\lambda$ and $\theta$. In our channel exchanging process, $\lambda$ is the weight of sparsity constraint (in Equation 4 of the main paper), and $\theta$ is the threshold for choosing close-to-zero scaling factors (in Equation 6 of the main paper). We conduct five experiments for each parameter setting. In the 1st and 3rd sub-figures, $\lambda$ ranges from $0.1 \times 10^{-3}$ to $30.0 \times 10^{-3}$, and $\theta$ is set to $10^{-2}$. In the 2nd and 4th sub-figures, $\theta$ ranges from $10^{-5}$ to $10^{-1}$, and $\lambda$ is set to $10^{-3}$. The task name is shown at the top of each sub-figure. The left y-axis indicates the metric, and the right y-axis indicates the proportion of channels that are lower than the threshold $\theta$, i.e. the proportion of channels that will be replaced. We observe that both hyper-parameters are not sensitive around their default settings ($\lambda = 10^{-3}$ and $\theta = 10^{-2}$).