[Reviews · NeurIPS 2020]

Review 1

Summary and Contributions: This paper examine the idea how to fuse multimodal inputs by interleaving channels of two networks by replacing dead-channels (i.e. when the BN param \gamma = 0) by the other network activations. After providing some theoretical insight, they evaluate their approach on semantic segmentation with a RGBD inputs and Image-to-Image translation. [Note: the reviewer is experienced in multimodal learning, and visually grounded language tasks. The reviewer is only familiar with the RGBD literature.]

Strengths: The current approach moves away from the classic late-fusion paradigm in deep learning (i.e. fusing modalities in a final neural block) by interleaving the channels of the different modalities at different levels. On a personal note, I think this is a promising line of research, and the proposed method is quite simple: 1) create sparse activation by using a L1 norm over the batchnorm scaling factor 2) replacing the activations. It is pleasant to see a simple and potentially working method. However, as discussed in the "Relation to Prior work", this line of research share many similarity with Conditional Batch Norm, and it is very surprising that no links are drawn between methods. After looking at the NYUDv2 literature, it seems that the table2 results are quite strong. Why are there two columns Commonly setting vs Same with our setting, while it is said l201 that you follow the standard setting.

Weaknesses: The authors compare their work to concatenation and alignment baselines. However, I am missing additional method-based related benchmarks. For instance, another mid-fusion baseline (or ablation) could be: - summing activation (only keep the second part of Eq6) - summing activation and adding a soft/hard gating mechanism - performing CBN \gamma = gamma_m1 + gamma(f(x_m2)) - turning the mixture of expert into a fusion block In the end, this paper shows the interest of performing non-naive mid-fusion, but some fundamental part of the approach remains obscure. Can we exchange channels without l1-normalization (e.g. thresholds) ? How does it compare with the modulation approach? Can we unshare the convnet? Why reglazing on half-of the params? As discussed below, the theoretical motivation has a few weaknesses, and it seems to not correlate with the experimental setting.

Correctness: In Table1: you show that sharing the convolutional parameters provide an important boost. In the studied tasks, it makes sense as RGB and Depth channels may share many similarities, and this can be done. What happens if you do not share the convnet but perform Channel exchanging? Do you observe an alignment of the channels? If yes, what is the final performance, if no, can you elaborate on this strong limitation. In Table4: how does it compare with concurrent methods of the literature? The author tried to provide a theoretical intuition of the proposed method. - Theroem1: I am not sure to understand the interest of this theorem. By construction, a lasso-regularisation over the parameters make the parameters sparse. Why demonstrating it? A simple reference to The Elements of Statistical Learning would have been enough. - Corrolarry1: I am very skeptic by both the underlying concept and the proof; Adding channel exchanging would reduce the loss. Why designing a specific f'_{1:M} is enough to prove the corollary. If we define f'_{1:M} to be a constant function equals to 1e6, then the loss should be higher. Besides, Table6 in the appendix shows that exchanging features on all the channels is less efficient than exchanging half-of-the channels. (Note that this is not due to the l1 regularisation as observed in the same table).

Clarity: Overall the paper is not hard to follow. My main concern is that there are multiple unjustified claim (l35-40), l60. I am quite concerned with the claim that the method can be extended to other modality l174. As language networks are based on RNN (with no batchnorm) and transformers, why do you think it would work?

Relation to Prior Work: There is a large portion of the multimodal learning literature based on modulation (Dumoulin et al. 2018) that is never mentioned in this paper. This is even more surprising as the authors refer to instance normalization, which was among the seminal work for modulation applied to style-transfer. Besides, Conditional Batch Norm (DeVries et al. 2017) have clear links and motivations with the current paper: this is critical missing reference. In CBN and its variant, the parameters of the batchnorm are conditioned on the other modalities. I would recommend updating the paper with the following references: - l34: early/late fusion is a pretty old concept (which is ill-defined in deep learning). You can find different definitions with: (Atrey et al.,2010; Bruni et al., 2014; Hall and Llinas, 1997; Snoek et al., 2005). The standard definition is often taken by Snoek et al. (2005). There have been new definition in the deep learning literature (Kiela 2017, Lazaridou et al. 2015, DeVries et al. 2017). In any case, two recent citations is not enough to define early/late fusion :) - Introduction 3rd paragraph: multiple claims are not supported by the literature, e.g, "aggregation-based fusion is prone to underestimating ... have been aggregated" is not justified. Same remark l39-40. - Related work: the authors speak about aggregated fusion and alignement based fusion. This taxonomy was defined by (Baltrušaitis et al. 2019), and it is clearly a missing citation - There exist other work that examines the impact and interest of batchnorm parameters,and how to handle dead channels such as (Wenqi et al. 2020), which is related to the current approach - the exchange of modality has also been explored with other modalities. cf Baltrušaitis et al. Section 7.1 paragraph2, but none of these methods are mentioned in the paper Overall, I strongly encourage the authors to frame their work in the current literature better, as several key references are missing. Dumoulin, Vincent, et al. "Feature-wise transformations." Distill 3.7 (2018): e11. C. G. M. Snoek, M. Worring, and A. W. M. Smeulders. Early versus late fusion in semantic video analysis. In Proceedings of the ACM International Conference on Multimedia, 2005 P. K. Atrey, M. A. Hossain, A. El Saddik, and M. S. Kankanhalli. Multimodal fusion for multimedia analysis: a survey. Multimedia systems, 16(6):345–379, 2010. D. L. Hall and J. Llinas. An introduction to multisensor data fusion. Proceedings of the IEEE, 85(1):6–23, 1997. E. Bruni, N.-K. Tran, and M. Baroni. Multimodal distributional semantics. Journal of Artificial Intelligence Research, 49: 1–47, 2014 D. Kiela. Deep embodiment: grounding semantics in perceptual modalities. Technical report, University of Cambridge, Computer Laboratory, 2017. A. Lazaridou, E. Bruni, and M. Baroni. Is this a wampimuk? cross-modal mapping between distributional semantics and the visual world. In Proceedings of the Annual Meeting of the Association for Computational Linguistics (ACL), 2014. H. de Vries, F. Strub, S. Chandar, O. Pietquin, H. Larochelle, and A. Courville. GuessWhat?! Visual object discovery through multi-modal dialogue. In Proceedings of IEEE Conference on Computer Vision and Pattern Recognition (CVPR), 2017 T. Baltrušaitis, C. Ahuja, and L.-P. Morency. Multimodal machine learning: A survey and taxonomy. IEEE Transactions on Pattern Analysis and Machine Intelligence, 41(2):423–443, 2019. Shao, Wenqi, et al. "Channel equilibrium networks for learning deep representation." ICML (2020). ----------------------------------- Thank you for this excellent rebuttal and the new baselines. I am stil not convinced that the method can be easily extended to other modalities (a simple MLP may not easily output a convnet activation), but the methods remain sound for RGB-D and related vision tasks. I thus updated the paper score.

Reproducibility: Yes

Additional Feedback: Small remarks: - Abstract: "Sufficient" ablation studies have also been carried out. "Sufficient" is quite subjective, and I would remove it - Eq6: theta -> 0


Review 2

Summary and Contributions: This paper presents a channel-exchanging-network, a parameter-free multi-modal fusion model, to dynamically exchange channel information between sub-networks of different modalities. Experiments are conducted on two tasks to validate the effectiveness of the proposed method.

Strengths: --The model is interesting to me. The importance of each channel is measured by the scaling factor of BN or IN and weak channels are replaced by the averaged channels of other modality, and effective fusion of different modalities is thus achieved. --The method has sufficient theoretical analysis and thorough experiments. --The model is likely to be of interest to a large proportion of the community.

Weaknesses: --Some experiments are missing. The comparison of the method against the random exchanging in ShuffleNet should be reported. In addition, if those unimportant channels are judged, you directly discard them. What are the advantages of the proposed method over this operation? The experiment should be conducted to show the effectiveness of channel exchanging over directly discarding the unnecessary channels. --In equation (6), the current channel is replaced with the mean of other channels if its scaling factor is smaller than a certain threshold. Why use the mean of other channels? Are there any other choices?

Correctness: Yes.

Clarity: Needs improvements.

Relation to Prior Work: Yes.

Reproducibility: Yes

Additional Feedback:


Review 3

Summary and Contributions: It has been observed that multi-modal fusion often does not lead to performance improvement with a simple fusion approach. The hypothetical reason for this non-trivial fusion issue is deemed due to the difficulty of balancing the trade-off between the multi-modal information. This submission proposes a novelty way of tackling this balancing problem in multi-modal fusion, i.e., channel exchange. The authors argue that this approach is a parameter-free approach to adjust the balance. In order to select the channels to be exchanged between two modalities, the authors measure channel importance by the magnitude of the Batch-Normalization scaling factor, denoted as \gamma, which is motivated by the channel pruning [38]. But the authors extend the understanding by Theorem 1. Also, Eq. (6) (channel exchange) is interesting in that, according to Corollary 1, there always exists a linear combination of other modality channels that decrease or at least unchange the loss (it is intuitive though). The proposed architecture is compact by sacrificing the applicability: the proposed method is only applicable to the problems where the input data shape of modalities have the same (compatible) form. This submission is a strong submission w.r.t. writing, technical contribution, and evaluation. =================Updated after rebuttal================== This reviewer has read all the reviews and the authors' rebuttal. This reviewer supports to accept this work. In the following, this reviewer has a few comments. While it was good to confirm that Theorem 1 still holds (I knew), this reviewer's point was not on the validity of Thoerem1, but on the gap between practice and theory. With \beta!=0, the probability that Theorem 1 holds is not necessarily with high probability. Thus, without \beta=0, the claim in L184-186 is not a guaranteed claim. This can be reflected in the final version. Also, this reviewer agrees with R1's comment on the related work and baselines. The authors are recommended to reflect the comments properly in the final version.

Strengths: - This submission address a multi-modal fusion balancing issue - The property of the algorithmic behaviors are adequately formalized - The performance is noticeably increased while keeping compact network architecture - The experiment results demonstrate the effectiveness of the proposed method adequately.

Weaknesses: - Theorem 1 is derived and interpreted under the assumption that \beta is ignorable (L178-179), which introduces the gap between Theorem 1 and practice. But the authors empirically show that the result still holds. (see the additional feedback below) - The proposed method is not applicable to a completely heterogeneous form of multi-modal data, e.g., image + audio.

Correctness: - Except the assumption (\beta = 0) and over-generalized interpretation to the practical case (\beta!=0), everything looks good.

Clarity: - The paper is well-motivated, and the description is clear.

Relation to Prior Work: - The baselines compared in this work are deemed to be sufficient.

Reproducibility: Yes

Additional Feedback: - L178-179, L186-188: Although the authors demonstrate the empirical result that many \gamma is stuck into 0 (a local minimum), the reason that \gamma is converged to zero during training (not at a convergence point) may not be necessary due to a small gradient if \beta is not zero. In an extreme case that \beta != non-zero, \gamma == zero, and x-mu != 0, then x' = \beta; thus, there always exists a descent direction dL/dx' that is non-zero. This implies that there is at least a descent direction to \gamma that is escapable from a basin of attraction. In short, Theorem 1 with the assumption of \beta == 0 introduces the gap between the theory and the actual implementation. In this regard, it would have been interesting to see the algorithmic behavior of the model implementation with no \beta. However, this cannot be the reason to discount this submission if the authors discuss this part properly in the paper. Anyway, resultantly, the result happens to be held, as shown in Fig. 5 in the supplementary material due to some other reasons. This reviewer would like to hear the authors' feedback on this part.


Review 4

Summary and Contributions: This paper proposes a new approach for  multimodal fusion. The method is parameter-free and only uses the coefficient of batch normalization (BN) as a criterion to replace some unimportant channels. The proposed architecture is relatively straightforward and composes BN as the importance measurement of each corresponding channel. It has better performance on semantic segmentation and image translation. In addition, it includes some theoretical analysis and nice ablation studies which help explain why it works.

Strengths: 1. The main contribution is the method on network pruning for multi-modal fusion, which allows channel-wise information to be exchanged between modalities but also retain intra-modal features. 2. The paper reinterprets modality-common and modality-specific patterns in the context of intra-modal and inter-modal information fusion. 3. The paper is demonstrated through well-designed experiments.

Weaknesses: 1. While the paper does achieve superior results, large chunks are devoted to highlighting BN and arguing that BN approaches can do better than other approaches. To this end, the novelty seems to be limited to putting the similar idea with [38]. 2. Authors make a point about exchanging channel — but exchanging channel is only achieved using BN. I am not really sure if such a contributionis incremental. In my opinion, the rationale of Theorem 1 are mostly based on [38], and Theorem 1 seems like a simple fact. 3. It would be better if the authors provide a figure of the percentage of exchanged channel for every layer, which will help readers better understand the model. 4. As stated in the paper, all modalities must be homogeneous.

Correctness: The claims and method are correct.

Clarity: The paper is well written while as a research paper, the contributions are not so significant.

Relation to Prior Work: The difference between the proposed method and prior work is clearly discussed.

Reproducibility: Yes

Additional Feedback: Overall, the idea is simple yet compelling. But I do have the following questions: 1. In eq(6), the current channel is replaced with the mean of other channels if the scaling factor is smaller than a certain threthod. If there are three modalities (represented by A, B, and C) and two of these (suppose they are B and C) are lower than the thredthod, the x of B is the mean of A and C, or just be replaced by A? And do all experimental settings specify two modalities? Does it make effect for three or more modalities? 2. I failed to run the segmentation code through your github link. The error is “CUDA out of memory”, but my GPU’s memory is 12G. Thus I suggest mentioning the GPU device type clearly in the paper. 3. Does the L1 penalty really work? Beacause I noticed that in Table 6 of the supplementary material, the mean IoUs shown in the third row are similar to those shown in the sixth row (46.2/38.4/48.0 vs. 46.0/38.1/47.7).

[Author Response · NeurIPS 2020]

Table A: More ablation studies on dataset NYUDv2 with ResNet101. All experiments use unshared BNs. Ensemble indicates using ensemble to mix predictions of RGB&Depth.

| Method | Convs | Ensemble | Mean IoU |
|---|---|---|---|
| Summing activation | Shared | ✓ | 48.1 (%) |
| Summing activation + soft gate | Shared | ✓ | 48.4 (%) |
| CBN | Unshared | ✓ | 48.3 (%) |
| CBN | Shared | ✓ | 48.9 (%) |
| Ours (concat-fusion block) | Shared | ✗ | 50.8 (%) |
| Ours | Shared | ✓ | **51.1** (%) |

Figure A: Percentage of exchanged channels on dataset Taskonomy.

Task: shading+texture→RGB
X-axis: Layer index
Y-axis: Channel number
▢ Not exchanged channels
▢ Exchanged channels of the **texture** modality
▢ Exchanged channels of the **shading** modality

We thank all reviewers for their constructive comments and address the reviewers' concerns point by point below.

**[To Reviewer #1:]** **Q1. More mid-fusion baselines:** Thanks for the suggestions. We have conducted extra ablation
studies raised by the reviewer in Table A. For the CBN strategy [DeVries et al. 2017] specifically, we modulate the BN
of one modality conditional on the other. In this sense, CBN performs cross-modal message passing via BN modulation,
which is clearly different from our method that directly exchanges channels between different modalities for fusion. As
expected, all ablations (including CBN) are inferior to our method, which again verifies the validity of the proposed
mechanism. Per the reviewer's concern, experiments on other datasets will also be added into the final version.

**Q2. More explanations on fundamental parts: 1.** We can **not** perform channel exchanging without $\ell_1$, as it enables
the discovery of unnecessary channels and comes as a pre-condition of Theorem 1. Naively exchanging, *e.g.*, 30%
channels only gets IoU 47.2. **2.** We provide the comparison with the modulation approach in Table A. **3.** We can conduct
channel exchanging upon unshared CNN, which has been evaluated in Table 6 (the 5th and 7th rows). Promisingly, its
full-channel case getting 49.1 still outperforms baselines in Table 2 under the same setting. We believe our method is
generally useful and channels are aligned to some extent given unshared CNN. **4.** We have discussed why regularizing
half of the channels is beneficial in L155-161 and provided its empirical evidence in Table 1 (the 5th and 6th rows).

**Q3. The rationality of our theorems: Theorem 1** is meaningful and crucial. Yes, $\ell_1$ makes the parameters sparse, but
it can not tell if each sparse parameter will keep small in training considering the gradient in Eq.(4). Conditional on BN,
Theorem 1 proves that $\gamma = 0$ is attractive, which is nontrivial and novel. **Corollary 1** states that $f'$ is more expressive
than $f$ when $\gamma = 0$, and thus the optimal $f'$ always outputs no higher loss, which, yet, is not true for arbitrary $f'$ (*e.g.*
$f' = 10^6$). Besides, Corollary 1 holds upon unshared CNN (see L191) and is consistent with Table 6 in the unshared
scenario (full-channel: 49.1 vs half-channel: 48.5), although full-channel exchanging is worse under the sharing setting.

**Q4. On recommended references:** We thank for the comment, will cite all the raised references (including the
modulation-based and CBN methods), and will refresh the references related to early/late fusion and the fusion
taxonomy. Our claims of "Introduction 3rd paragraph" are intuitively supported by the observations from Figure 1,
namely, the aggregation-based fusion integrates multiple networks into one single branch and the alignment-based
fusion conducts an indirect way of fusion by training. We will further clarify this in the final version.

**Q5. Other comments: 1.** In L201, "the standard setting" refers to train/test split, while "commonly/same with our
setting" is on architecture design, as already described in L236-238. **2.** We agree our current version is inapplicable for
heterogeneous fusion (*e.g.* CNN+RNN). That is why we, explicitly and honestly, point it out in L170-176. Given the
effectiveness on homogeneous fusion, we conjecture that adding extra homogeneous MLP layers upon each CNN/RNN
enables channel (or neural unit) exchanging for heterogeneous fusion, which will be left for future exploration.

**[To Reviewer #2:]** **Q1. More experiments:** We have carried out random exchanging like ShuffleNet or directly
discarded unimportant channels without channel exchanging on NYUDv2, the IoUs of which are 46.8 and 47.5,
respectively. Both are much worse than our method (51.1). We are willing to add these ablations into the final version.

**Q2. Why use the mean:** We have tried weighted summation of channels, but do not observe clear improvement (FID
from 60.90 to 60.94 on the translation task Depth+Normal+Texture→RGB). We conjecture that $\gamma$ of each modality in
Eq. (6) already contributes to the weights, thus the simple and parameter-free mean delivers promising performance.

**[To Reviewer #3:]** We thanks for the positive comments, and are sorry for the unclear presentations on Theorem 1.
We would like to highlight that our conclusion always holds regardless of the value of $\beta$. Our proof (in appendix) is
interested in the gradient $dL/d\gamma$ only around $\gamma = 0$ (not other points) and finds that the probability of staying around
$\gamma = 0$ is large and hits the Gaussian peak, even for the extreme case ($\beta$ or $dL/d\mathbf{x}'$ is non-zero) raised by the reviewer.
By mentioning $\beta = 0$ in L179, we initially tend to simplify the analysis of Corollary 1. Yet, our current proof of
Corollary 1 does not require this assumption, and we will remove $\beta = 0$ to avoid confusion in the final version.

**[To Reviewer #4:]** **Q1. Novelty and contributions compared to [38]:** Our paper is inspired but differs from [38] in
two aspects: **1.** while [38] aims at model compression, our paper attempts to conduct multimodal fusion. Thus upon
[38], we employ channel exchanging for message passing, which is simple yet effective and novel compared to current
fusion methods. **2.** Theorem 1 is related to [38], but [38] never confirms that unnecessary channels during training can
hardly recover, and we rigorously prove it by exploring the behaviors of the $\ell_1$ norm and BN layers.

**Q2. More discussions: 1.** Thanks for the reminding, and we have reported the percentage of exchanged channel for
every layer in Figure A. It shows that upper layers are more active in exchanging. **2.** Please refer to Q5.2, R#1 for the
discussion on heterogeneous fusion. **3.** Regarding the three-modal example (A, B, and C) raised by the reviewer, the
$\ell_1$ norm is conducted on disjoint channels of different modalities. It means in Eq.(6) that for each channel, only one
modality will meet the criterion (scaling factor lower than the threshold). This setting will avoid unavailing exchanging
as already described in L160-161. **4.** Not all experiments specify two modalities. We have tested the effectiveness of
our method with modalities more than two in Table 5 (also Table 7, 8 in appendix), where our method still makes effect.

**Q3. Other comments:** We use V100 with GPU memory 32G, and the segmentation needs 20G. We will detail the
GPU type in the final version. Applying the $\ell_1$ norm alone delivers no improvement, but it is necessary for the discovery
of unnecessary channels and channel exchanging (see also our response in Q2.1, R#1).

[Meta-Review · NeurIPS 2020]

The paper presents an interesting method for multimodal fusion based on feature channel exchanging across modalities. All reviewers recommend acceptance. The AC agrees with the consensus reached by the reviewers and request the authors to improve the related work discussion as pointed out by R1 and add the discussion in the rebuttal to the final version of the paper.